# Efficient Synthesis of Furfuryl Alcohol from Corncob in a Deep Eutectic Solvent System

**Lizhen Qin** [1], **Junhua Di** [2] **and Yucai He** [2,*] ●

1   School of Chemistry and Chemical Engineering, Jiangsu University of Technology, Changzhou 213001, China
2   School of Pharmacy, Changzhou University, Changzhou 213164, China
*   Correspondence: heyucai2001@163.com or yucaihe@cczu.edu.cn

**Abstract:** As a versatile and valuable intermediate, furfuryl alcohol (FOL) has been widely used in manufacturing resins, vitamin C, perfumes, lubricants, plasticizers, fuel additives, biofuels, and other furan-based chemicals. This work developed an efficient hybrid strategy for the valorization of lignocellulosic biomass to FOL. Corncob (75 g/L) was catalyzed with heterogenous catalyst Sn-SSXR (2 wt%) to generate FAL (65.4% yield) in a deep eutectic solvent ChCl:LA–water system (30:70, *v/v*; 180 °C) after 15 min. Subsequently, the obtained FAL liquor containing FAL and formate could be biologically reduced to FOL by recombinant *E. coli* CF containing aldehyde reductase and formate dehydrogenase at pH 6.5 and 35 °C, achieving the FOL productivity of 0.66 g FOL/(g xylan in corncob). The formed formate could be used as a cosubstrate for the bioreduction of FAL into FOL. In addition, other biomasses (e.g., sugarcane bagasse and rice straw) could be converted into FOL at a high yield. Overall, this hybrid strategy that combines chemocatalysis and biocatalysis can be utilized to efficiently valorize lignocellulosic materials into valuable biofurans.

**Keywords:** furfuryl alcohol; furfural; biomass; solid acid; deep eutectic solvent; reductase

## 1. Introduction

To alleviate the increasing reliance on non-renewable fossil resources, great research efforts have been devoted worldwide to exploring sustainable processes for manufacturing valuable biobased chemicals and biofuel molecules from available, low-cost, abundant, and renewable resources [1,2]. Furfuryl alcohol (FOL) is known as a primary derivative of furfural (FAL), which can be utilized in the production of various plasticizers, furan resins, lubricants, rubbers, fibers, vitamin C, lysine, dispersing agents, and biofuels [3,4]. Approximately 62% of FAL has been utilized for FOL production [5]. Industrially, FOL is mainly prepared from FAL via the gas- or liquid-phase hydrogenation approach [6,7]. However, these two chemical dehydrogenation processes require high energy consumption or noble catalyst, accompanied by undesired byproducts [8].

In recent years, the biological reduction has increasingly been investigated as a promising manufacturing technique for FOL production. Distinct from chemical dehydrogenation, biological reduction has gained great interest due to mild performance conditions, attractive selectivity, suitable catalytic activity, and eco-friendliness [9]. In fact, the cofactor NAD(P)H is required by about 80% of oxidoreductases. Presently, the coupling of aldehyde reductase (ALDH) and dehydrogenase for recycling NAD(P)H has been efficiently utilized for synthesizing alcohols. Formate dehydrogenase (FDH) and glucose dehydrogenase (GDH) have been commonly employed to regenerate NAD(P)H accompanied by consumed substrates (formate and glucose) during bioreduction [10,11]. GDH can convert glucose to form gluconic acid. In contrast, FDH may catalyze formate to produce $CO_2$. Notably, the generated gluconic acid needs to be removed by an additional separation process. While the small amount of $CO_2$ generated may evaporate from the reaction system [12]. Obviously, co-expression of FDH and NADPH-dependent ALDH is a suitable strategy for the bioreduction of FAL into FOL.

Lignocellulosic biomass (LCB), which is composed of cellulose, lignin, and hemicellulose, is regarded as a very important alternative energy and carbon-based chemical source [13] due to its availability on Earth. Hemicellulose has high potential since its hydrolysis and dehydration can produce FAL, which is considered a building block molecule [14]. Homogeneous acid catalysts (e.g., mineral acids, organic acids, and their mixtures) have been utilized to convert biomass, xylan, and xylose into FAL [15]. Distinct from homogeneous acid catalysts, heterogeneous catalysts have high thermostability and suitable recyclability [16–18]. Besides the efficiency of chemocatalyst for high productivity and high selectivity in the production of FAL, solvent also has a key role in the enhancement of FAL productivity [19]. Recently, deep eutectic solvent (DES), which is prepared by mixing a hydrogen bond donor (HBD) and a hydrogen bond acceptor (HBA) [20], has been used as green reaction solvents for the application of biocatalysis and chemocatalysis [21–23]. It is also utilized in the extraction of polyphenols from biomass [24], the delignification of biomass [25,26], and the enantiodiscrimination of chiral compounds [27]. To reduce the dosage of DES, a mixture of DES and water has been utilized [28,29]. It has gained great interest to utilize in the efficient production of FAL in a green water-DES system.

Tin-based heterogeneous catalyst Sn-SSXR can be used for the efficient conversion of biomass into FAL [30]. Cells of recombinant *E. coli* CF containing FDH and ALDH can be employed to transform FAL into FOL using formate as a cosubstrate [31]. DES ChCl:LA has the ability to remove lignin and catalyze biomass into FAL, along with solvent recyclability [32]. In our study, one hybrid conversion of biomass to FOL was carried out in a tandem chemoenzyamtic reaction with chemocatalysis and biocatalysis in a DES ChCl:LA–water system (Scheme 1). Various chemocatalytic and biocatalytic reaction parameters were examined on the catalytic efficiency. An efficient bioreduction of corncob-derived FAL into FOL using corncob-derived formate as cosubstrate was attempted to be conducted in a DES ChCl:LA–water system. One efficient and sustainable transformation of biomass into high-value chemicals was developed in the aqueous media.

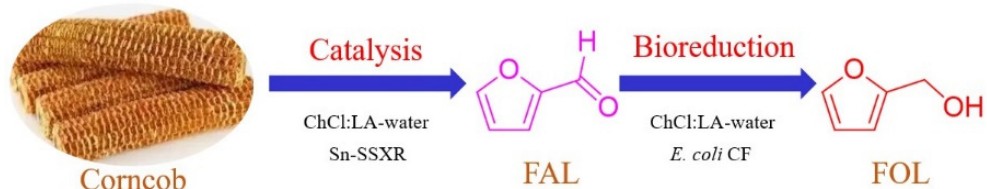

**Scheme 1.** Outline of the synthetic pathway.

## 2. Materials and Methods

### 2.1. Materials and Reagents

ChCl (≥99%), lactic acid (LA) (≥99%), Sodium formate (HCOONa) (≥99%), $NADP^+$ (≥99%), furfural (FAL) (≥99%), furfuryl alcohol (FOL) (≥99%) and other chemicals were bought from Sinopharm Group Chemical Reagent Co., Ltd. (Shanghai, China).

### 2.2. Synthesis of FAL from Biomass

DES ChCl:LA was prepared from ChCl and lactic acid (LA), as previously reported [26]. Heterogeneous catalyst Sn-SSXR was prepared as previously reported [30]. A total of 3.0 g biomass powder (e.g., corncob, sugarcane bagasse, and rice straw) to FAL with Sn-SSXR (0.5–5 wt%) at 160–180 °C for 5–40 min in an autoclave reactor (Dantu Huanqiu Electrical Instrument Co., Zhenjiang, China) containing 40 mL ChCl:LA–water system. The yield of FAL was calculated as below:

$$\text{FAL yield } (\%) = \frac{\text{FAL formed} \times 0.88}{\text{Xylan in biomass } (\text{g})} \times \frac{150}{96} \times 100\% \tag{1}$$

### 2.3. Whole-Cell Bioreduction of FAL into FOL

Cells of recombinant *E. coli* CF containing FDH and ALDH were cultivated and harvested as previously reported [31]. To test the effects of NADP$^+$ dose, biocatalytic temperature, and biocatalytic pH on the bioreduction activity, biotransformation of diluted FAL liquor containing 100.0 mM FAL and 103.2 mM formate to FOL by CF whole cells (0.01–0.2 g/mL, wet weight) was carried out in DES-water containing NADP$^+$ (0–3 mmol NADP$^+$/mol FAL) at pH 6.0–8.0 and 25–45 °C in ChCl:LA–water (30:70, *v/v*).

The FOL yield was defined as follows:

$$\text{FOL yield } (\%) = \frac{\text{FOL formed (mM)}}{\text{Initial FAL (mM)}} \times 100 \qquad (2)$$

### 2.4. Analytical Method

Using the mixture of 20 v% methanol, 80 v% water, and 0.10 wt% trifluoroacetic acid as mobile phase (flow rate 0.8 mL/min), the concentrations of FAL and FOL were quantified with LC-2030C HPLC (3D SHIMADZU, Kyoto, Japan) using Athena C18 column (4.6 × 250 mm, 5 μm). FAL and FOL were observed at 254 and 210 nm, respectively.

## 3. Results and Discussion

### 3.1. Synthesis of FAL from Corncob in ChCl:LA–Water

Various catalytic parameters (e.g., solvent content, solid acid loading, performance temperature, duration, etc.) have a profound influence on the FAL generation [33–35]. Using Sn-SSXR (1–5 wt%) as a heterogeneous catalyst, corncob powders were used as feedstocks for the preparation of FAL in DES ChCl:LA–water (0:100–50:50, *v/v*) at 160–190 °C for 5–40 min. By raising DES ChCl:LA dose from 0 to 30 vol% at 180 °C for 15 min, FAL yield was rose from 41.0% (69.7 mM) to 65.4% (126.3 mM) in the presence of Sn-SSXR (2.0 wt%) (Figure 1a). As the volumetric ratio of ChCl:LA to water was raised from 30:70 to 50:50, FAL yield began to decline gradually. In the presence of ChCl:LA (50 vol%), FAL yield reached 47.2%. An excessive water loading in ChCl:LA–water could compete to form hydrogen-bonding with ChCl and displace LA, which would result in the weak hydrogen-bonding interactions occurring between LA and ChCl. Obviously, the optimal DES ChCl:LA dose was 30 vol%, and ChCl:LA–water (30:70, *v/v*) was utilized as an optimal medium for transforming corncob into FAL. In ChCl:LA–water (30:70, *v/v*) at 170 °C, FAL yield comparatively rose when the Sn-SSXR dose increased from 1.0 to 2.0 wt%. The maximum FAL yield reached 65.4% using 2.0 wt% of Sn-SSXR after 20 min (Figure 1b). When the Sn-SSXR amount was raised from 2.0 wt% to 5.0 wt%, the FAL yield had no obvious change. An excessive Sn-SSXR loading could not increase the acidity of the reaction system, which did not accelerate the generation of FAL [36]. Therefore, the appropriate Sn-SSXR dose was chosen as 2.0 wt%.

The catalytic temperature (160, 170, 180, and 190 °C) and catalytic duration (5–50 min) had a profound influence on the formation of FAL (Figure 2). Lower catalytic reaction temperature might not provide enough energy to dehydrate xylose into FAL by using Sn-SSXR as a catalyst. While a higher performance temperature might accelerate the FAL degradation, resulting in a decreased FAL yield due to the formation of undesirable byproducts [37]. The highest FAL yield reached 65.4% by using corncob (75.0 g/L) as feedstock in the presence of Sn-SSXR (2.0 wt%) as a chemocatalyst in ChCl:LA–water (30:70, *v/v*; 180 °C) for 15 min. The formed FAL liquor contained FAL (126.3 mM), 5-HMF (3.1 mM), formic acid (130.4 mM) glucose (13.7 mM), *D*-xylose (31.2 mM), arabinose (3.6 mM), and levulinic acid (0.92 mM).

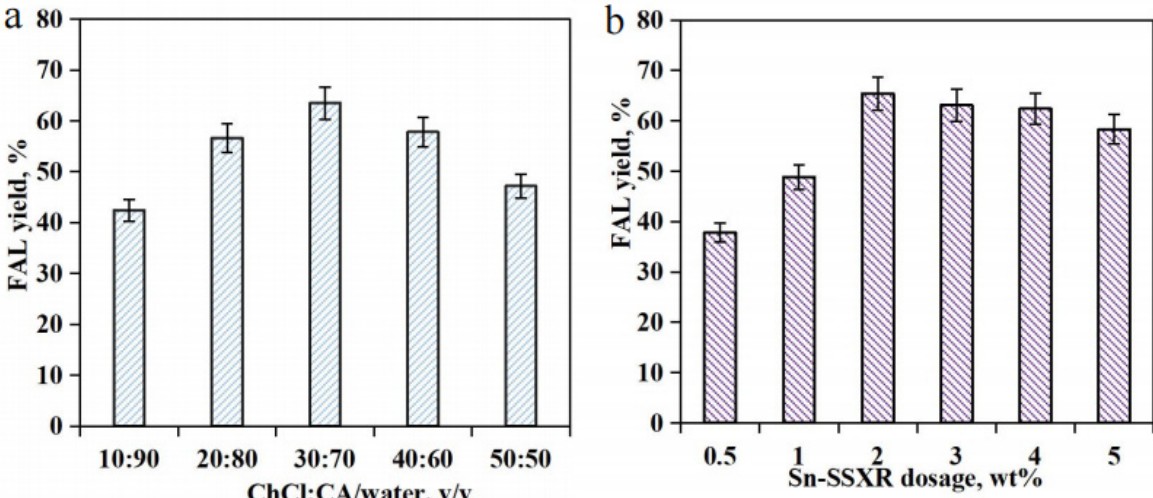

**Figure 1.** Effects of ChCl:LA loading (0–50 vol%) on the FAL yield (corncob 75 g/L, Sn-SSXR 2 wt%, 180 °C, 15 min) (**a**); Effects of Sn-SSXR dosage (0.5–5 wt%) on the FAL production in ChCl:LA–water (30:70, *v/v*) (corncob 75 g/L, 180 °C, 15 min) (**b**).

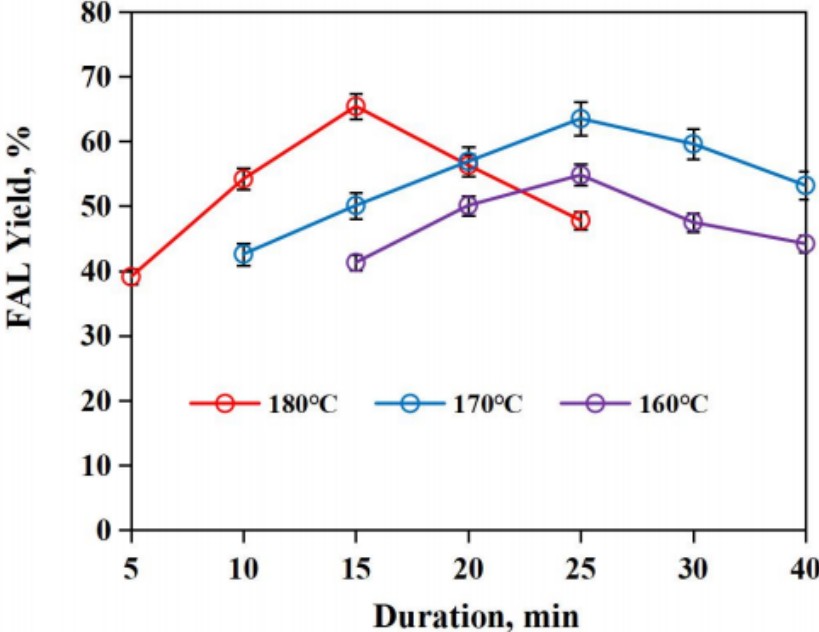

**Figure 2.** Effects of performance temperature (160, 170, and 180 °C) and duration (5–40 min) on the FAL yield in ChCl:LA–water (30:70, *v/v*) (CS 75 g/L, Sn-SSXR 2 wt%).

Tin-based heterogeneous catalysts (e.g., $SO_4^{2-}$/$SnO_2$-diatomite, $SO_4^{2-}$/$SnO_2$-kaoline, $SO_4^{2-}$/$SnO_2$-argil, $SO_4^{2-}$/$SnO_2$-CS, Sn-ZRD) could be utilized for efficient conversion of biomass or xylose into FAL (Table 1). Compared to Sn-SSXR, these three heterogeneous catalysts, including $SO_4^{2-}$/$SnO_2$-kaoline, $SO_4^{2-}$/$SnO_2$-argil, and Sn-ZRD, gave a lower titer of FAL. $SO_4^{2-}$/$SnO_2$-diatomite (3.6 wt%) could catalyze 75 g/L of corncob into FAL in a slightly high yield (68.9%) in a γ-GVL–water (6:4, *v/v*) system containing 15 g/L $ZnCl_2$ at 170 °C after 30 min. However, high-loading γ-valerolactone and $ZnCl_2$ were used. $SO_4^{2-}$/$SnO_2$-CS (1.2 wt%) could dehydrate xylose (75 g/L) into 200 mM FAL in ChCl:EG–water (5:95, *v/v*) at 185 °C within 20 min. However, FAL was obtained in a low yield (35.7%). Clearly, this established catalytic process by using Sn-SSXR as a chemocatalyst had a potential application for FAL production.

The reusability of chemocatalysts and reaction media has a vital role in the cost-effective manufacture of FAL [38,39]. After each reaction, FAL was extracted from FAL liquor three times using ethyl acetate by mixing in equal volumes. Furthermore, the reaction medium ChCl:LA–water was separated from solid samples (Sn-SSXR and biomass residues). After the solid samples were calcined in a muffle furnace (550 °C) for 4 h, the obtained heterogeneous catalyst regeneration was carried out by sulfonating the recycled catalysts before each batch of use. In this work, recovered and repeated reuse of heterogeneous Sn-SSXR and DES ChCl:LA was carried out for six batches. As depicted in Figure 3, FAL yield reached 65.4% in the first batch. In the second and third runs, a slight decrease in FAL yield was found. From the fourth to seventh run, FAL yield decreased gradually. At the 7th run, the FAL yield reached 34.1%. The reused Sn-SSXR and DES ChCl:LA–water had high stability for the conversion of biomass to FAL in ChCl:LA–water. The recovery and reuse of the catalytic system could relieve the economic burden and reduce environmental pollution and has a high application potential. The issue of DES recycling is a key factor in green chemistry. An efficient strategy for the recovery and reuse of DES ChCl:LA is in progress.

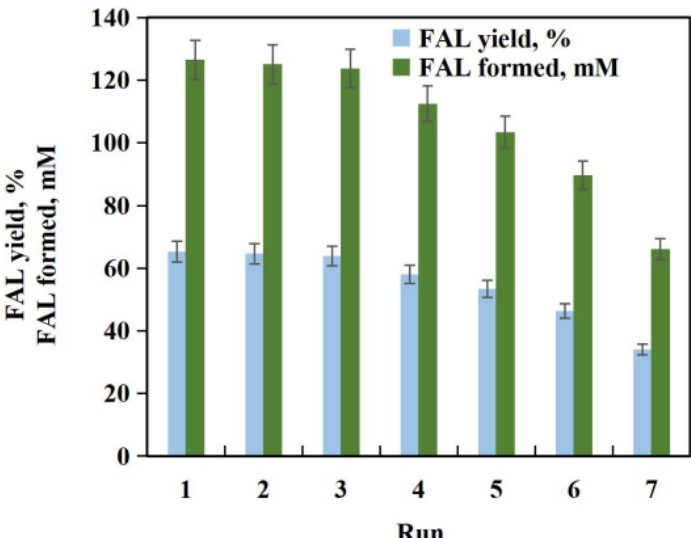

**Figure 3.** Recyclability of DES and heterogeneous catalyst. (40 mL ChCl:LA–water, 3 g corncob, hydrolysate, Sn-SSXR 2 wt%, 180 °C, 15 min).

**Table 1.** Related works about synthesis of FAL from biomass or xylose by using tin-based solid acid.

| DES | Reaction Conditions | Feedstock | FAL Concentration | FAL Yield | Ref. |
|---|---|---|---|---|---|
| $SO_4^{2-}$/$SnO_2$-diatomite (3.6 wt%) | γ-Valerolactone–water (6:4, $v/v$), 15 g/L $ZnCl_2$, 170 °C, 30 min | Corncob (75 g/L) | - | 68.9% | [16] |
| $SO_4^{2-}$/$SnO_2$-kaoline (3.5 wt%) | Toluene–water (1:2 $v/v$), 10 mM OP-10, 170 °C, 30 min | Corncob-derived xylose (20.1 g/L) | 50.5 mM | 74.3% | [4] |
| $SO_4^{2-}$/$SnO_2$-argil (3.6 wt%) | Water, 180 °C, 20 min | Xylose (20 g/L) | 107.6 mM | 57.1% | [9] |
| Sn-ZRD (3.6 wt%) | Water, 170 °C, 30 min | Dewaxed corncob (75 g/L) | 90.3 mM | - | [34] |
| $SO_4^{2-}$/$SnO_2$-CS (1.2 wt%) | ChCl:EG–water (5:95, $v/v$), 185 °C, 20 min | Xylose (75 g/L) | 200 mM | 35.7% | [39] |
| Sn-SSXR (2.0 wt%) | ChCl:LA–water (30:70, $v/v$), 180 °C, 15 min | Corncob (75 g/L) | 126.3 mM | 65.4% | This study |

### 3.2. Synthesis of FOL from Biomass-Derived FAL via Bioreduction in ChCl:LA–Water

It is well known that biocatalysis offers a high selectivity and eco-friendly approach for transforming carbonyl/aldehyde compounds into valuable alcohols [40,41]. The whole cells of *E. coli* CF harboring ALDH and FDH could be employed to biologically reduce diluted corncob-derived FAL (100 mM) to FOL in ChCl:LA–water. By increasing the cell dosage from 0.01 to 1.0 g/mL, the reaction rate increased from 2.1 to 7.0 μmol/min (Figure 4a). When the dosage of CF cells was increased from 0.12 to 0.2 g/mL, the reaction rate dropped slightly. To efficiently reduce FAL, it was necessary to obtain the optimum $NADP^+$ dose. When the molar ratio of $NADP^+$-to-FAL was increased from 0 to 1:1 (μmol:mmol), the biocatalytic activity increased greatly (Figure 4b). In contrast, the activity dropped slightly by increasing the molar ratio from 1:1 to 2:1 (μmol:mmol). Upon raising this ratio from 2.5:1 to 3:1 (μmol:mmol), the activity decreased obviously. Thus, the appropriate cell and $NADP^+$ dosages were 0.1 g/mL and 1.0 μmol $NADP^+$/mmol FAL, respectively.

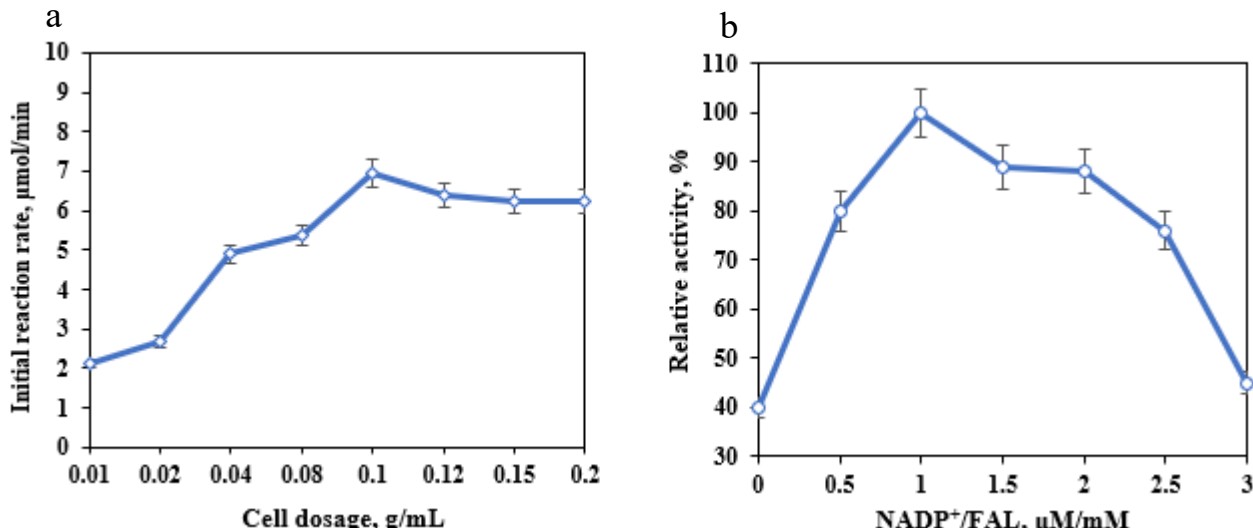

**Figure 4.** Effects of cell dosage (**a**) and $NADP^+$ dose (**b**) on FAL-reducing activity in ChCl:LA–water (30:70, *v/v*).

It was known that medium pH and biocatalytic temperature could obviously influence the FAL-reducing activity [42,43]. In ChCl:LA–water (30:70, *v/v*), the effects of performance temperature (25–45 °C) and medium pH (6.0–8.0) on FAL-reducing activity were tested using CF whole-cell (50 g/L, wet weight) as biocatalyst. The FAL-reducing activity clearly increased with the increase in bioreduction temperature from 25 to 35 °C (Figure 4a). Within 35 °C, higher bioreduction temperature accelerated the biocatalytic activity at pH 6.5. Over 35 °C, higher performance temperature decreased the FAL-reducing activity due to thermal inactivation. By increasing the medium pH from 6.0 to 6.5 at 35 °C, FAL-reducing activity was raised (Figure 4b). As the medium pH was increased from 6.5 to 8.0, the FAL-reducing activity dropped gradually. Clearly, the biocatalytic activity reached the maximum value at pH 6.5 (Figure 5a) and 35 °C (Figure 5b). Distinct from chemo-selective reduction [44,45], bioreduction is regarded as an eco-friendly route for the production of FOL under the eco-friendly performance condition [46,47].

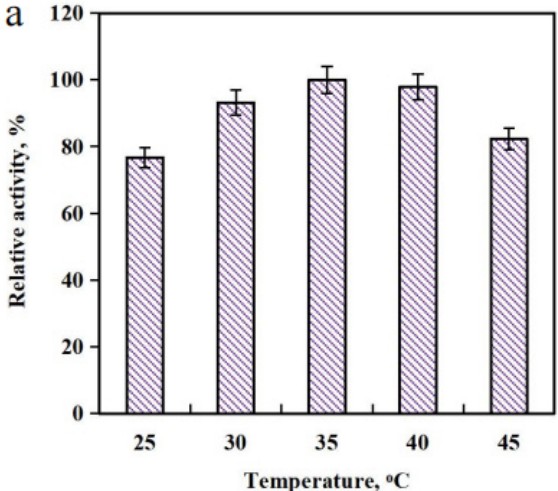

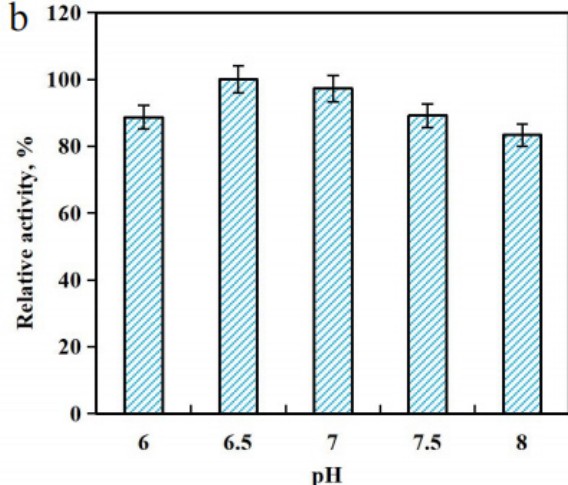

**Figure 5.** Effects of performance temperature (25–45 °C) (**a**) and medium pH (6.0–8.0) (**b**) on FAL-reducing activity in ChCl:LA–water (30:70, *v/v*).

### 3.3. Transformation of Corncob to FOL via Chemoenzymatic Approach in ChCl:LA–Water

Chemoenzymatic catalysis by bridging chemocatalysis and biocatalysis has been developed for preparing a series of valuable biobased chemicals from lignocellulosic materials [35,42]. In ChCl:LA–water (30:70, *v/v*), corncob (3.0 g, 75 g/L) was tandemly converted into FOL with Sn-SSXR chemocatalyst and CF cell biocatalyst. At 180 °C, Sn-SSXR (2 wt%) converted corncob to 126.3 mM FAL in a yield of 65.4% after 15 min. This FAL liquor, which was regulated to pH 6.5, was biologically reduced to FOL with CF cells (0.1 g/mL) in ChCl:LA–water (30:70, *v/v*). Time courses for bioreduction of dilute FAL (110.0 mM) were monitored. Within 165 min, FAL concentrations dropped quickly (Figure 6). In ChCl:LA–water, no significant inhibition was observed. Bioreduction for 165 min, FOL concentrations reached 110.0 mM, FAL could be fully converted into FOL, achieving the productivity of 0.66 g FOL/g xylan.

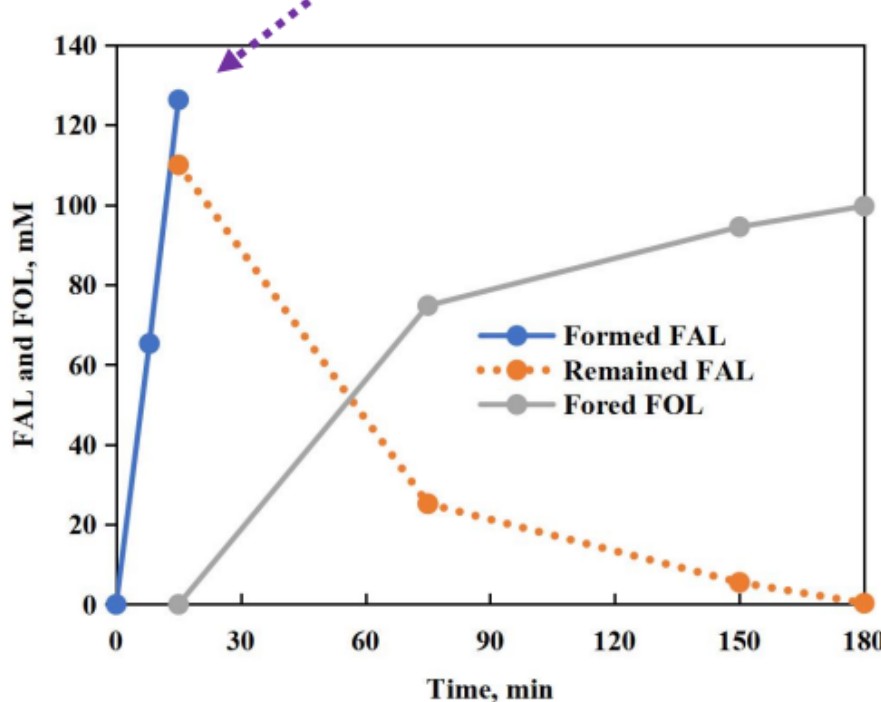

**Figure 6.** Tandem catalysis of corncob to FOL by bridging chemocatalysis with Sn-SSXR and biocatalysis with CF cell in DES ChCl:LA–water (arrow: pH adjustment and CF cell addition).

Furthermore, three biomasses (corncob, sugarcane bagasse, and rice straw) were chemoenzymatically catalyzed into FOL in a tandem reaction with Sn-SSXR and CF cell in ChCl:LA–water (30:70, $v/v$). Chemocatalysis with Sn-SSXR (2.0 wt%) could catalyze corncob, sugarcane bagasse, and rice straw (3.0 g, 75 g/L) into 126.3, 101.4, and 95.9 mM FAL at 180 °C after 15 min (Table 2). The FAL yields were obtained as follows: $Yield_{(Corncob)}$ = 65.4% > $Yield_{(Sugarcane\ bagasse)}$ = 60.3% > $Yield_{(Rice\ straw)}$ = 58.2%. Different biomasses with different content of xylan were used as substrates to produce FAL. Clearly, corncob was a suitable feedstock for FAL production due to its high xylan content and FAL yield. The formed FAL liquors, which were adjusted to pH 6.5, could be biologically reduced to FOL by CF cells (0.1 g/mL) in ChCl:LA–water (30:70, $v/v$). Bioreduction for 165 min, FAL could be fully converted into FOL, achieving the productivity of 0.59–0.66 g FOL/g xylan (Table 2). FAL could be obtained from the hydrolysis, dehydration, and cyclization of xylan in biomass. Hence, the titer of FAL derived from lignocellulosic biomass was mainly based on the xylan content in biomass. The biomass-derived FAL could be fully reduced to FOL by CF cells. Hence, corncob gave a high productivity of FOL via chemoenzymatic approach in ChCl:LA–water (30:70, $v/v$).

**Table 2.** Chemoenzymatic conversion potency toward different biomass in ChCl:LA–water.

| Biomass | Xylan Content, wt% | FAL, mM (Yield, %) [a] | FOL Productivity, g FOL/g Xylan [b] |
|---|---|---|---|
| Corncob | 34.0 | 126.3 (65.4) | 0.66 |
| Sugarcane bagasse | 29.6 | 101.4 (60.3) | 0.61 |
| Rice straw | 29.0 | 95.9 (58.2) | 0.59 |

[a] Various biomasses (75 g/L) were used as feedstocks for the production of FAL in 15 min at 180 °C in ChCl:LA–water (30:70, $v/v$) by Sn-SSXR (2.0 wt%). [b] The generated FAL was biologically reduced to FOL by supplementary CF cells (0.1 g/mL) in ChCl:LA–water (pH 6.5) at 35 °C for 165 min.

Lignocellulosic biomass is known as an available, abundant, low-cost, and renewable bioresource [48,49], which can be widely utilized for manufacturing value-added compounds [50,51]. DES has received considerable interest as a reaction solvent for chemocatalysis and biocatalysis because of its low toxicity, suitable reusability, high thermostability, and ease of preparation [52–54]. In this work, DES ChCl:LA–water (30:70, $v/v$) was established. Catalyst Sn-SSXR could be utilized to efficiently transform biomass into FAL in DES ChCl:LA–water, accompanied by the formation of formate. Recombinant *E. coli* CF containing ALDH and FDH could efficiently transform biomass-derived FAL containing formate. Formate could be used as a co-substrate for transforming FAL to FOL. Distinct from the chemical hydrogenation, biological reduction with CF whole-cell biocatalysts had higher FAL-reducing activity for efficiently transforming biomass-derived FAL to FOL under the ambient performance condition. The tandem catalysis by bridging chemocatalysis and biocatalysis was successfully constructed for the sustainable valorization of biomass into FOL in DES ChCl:LA–water (Scheme 2).

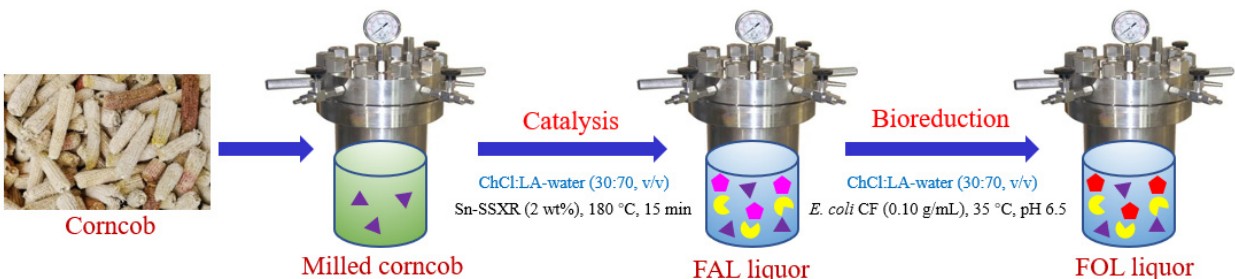

**Scheme 2.** Scheme for tandem catalysis of corncob to FOL in DES ChCl:LA–water.

## 4. Conclusions

FOL is one kind of furan-based chemical, which is primarily utilized for the production of furan resins that are widely used in thermoset polymers, coatings, plastics, adhesives, cements, and biofuels. It can be synthesized into a series of valuable furans. In industry, FOL can be prepared via chemical hydrogenation of FAL under high pressure of $H_2$ or through the disproportionation via the Cannizaro reaction in an alkali system. Notably, bioreduction of FAL into FOL can be conducted under mild conditions. In addition, the hemicellulose components in lignocellulosic biomass can be hydrolyzed into pentose (mainly *D*-xylose) in the acidic condition, and the formed *D*-xylose can be further dehydrated into FAL. To convert available, renewable, and abundant lignocellulose into FOL, chemoenzymatic conversion by bridging chemocatalysis and bioreduction can be attempted in biorefinery processes.

This study provided a novel route for efficient conversion of corncob into FOL in a cascade reaction with solid acid Sn-SSXR and whole-cell of recombinant *E. coli* CF. Firstly, corncob was catalyzed with Sn-SSXR into FAL (126.3 mM) in ChCl:LA–water (30:70, *v/v*), and the formed FAL solution containing FAL and formate were biologically transformed into FOL with recombinant *E. coli* CF harboring ALDH and FDH under the mild bioreaction conditions. In addition, other biomasses were also utilized as feedstocks for the effective production of FOL. Thus, one efficient tandem conversion by bridging chemocatalysis and mild biocatalysis was successfully developed for the sustainable conversion of biomass into FOL.

**Author Contributions:** Conceptualization, Methodology, and Writing—original draft, L.Q. and J.D.; Data curation, Software, Supervision, Review, and Revising manuscript, Y.H. All authors have read and agreed to the published version of the manuscript.

**Funding:** This research received no external funding.

**Institutional Review Board Statement:** Not applicable.

**Informed Consent Statement:** Not applicable.

**Data Availability Statement:** Not applicable.

**Acknowledgments:** The authors thank the Jiangsu Key Laboratory of Advanced Catalytic Materials and Technology (Changzhou University) for kindly donation of strain recombinant *E. coli* CF.

**Conflicts of Interest:** The authors declare no conflict of interest.

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
