# Peer review of "Efficient Synthesis of Furfuryl Alcohol from Corncob in a Deep Eutectic Solvent System"

_processes, doi:10.3390/pr10091873_

Round 1
Reviewer 1 Report
The authors reported the bioreduction of biomass-derived Furfural into Furfuryl alcohol. The submission can be accepted after major revision considering the following points:
1. The title should be revised to be short, precise, and informative. Abbreviations should be avoided.
2. The novelty should be highlighted.
3. A scheme showing the process of the material production and applications should be added.
4. A comparison with previously published methods and materials should be discussed and summarized in Tables.
5. Mass spectra of the reaction crude showing the furfural and their reduced product should be added.
6. The language should be revised and typos should be corrected.
Author Response
The authors reported the bioreduction of biomass-derived Furfural into Furfuryl alcohol. The submission can be accepted after major revision considering the following points:
- The title should be revised to be short, precise, and informative. Abbreviations should be avoided.
Response: Thanks for the good suggestion. The title was revised ad shortened as below:
Efficient Synthesis of Furfuryl Alcohol from Corncob in a Deep Eutectic Solvent System
- The novelty should be highlighted.
Response: Thanks for the good suggestion.
In “1. Introduction”, the novelty was emphasized as below:
Tin-based heterogeneous catalyst Sn-SSXR can be used for efficient conversion of biomass into FAL [30]. Cells of recombinant E. coli CF containing FDH and ALDH can be employ to transform FAL into FOL using formate as cosubstrate [31]. DES ChCl:LA has the ability to remove lignin and catalyze biomass into FAL along with solvent recyclability [32]. In our study, one hybrid conversion of biomass to FOL was carried out in a tandem chemoenzymatic reaction with chemocatalysis and biocatalysis in a DES ChCl:LA-water system. Various chemocatalytic and biocatalytic reaction parameters were examined on the catalytic efficiency. An efficient bioreduction of corncob-derived FAL into FOL using corncob-derived formate as co-substrate was attempted to conduct in DES ChCl:LA-water system. One efficient and sustainable transformation of biomass into high-value chemicals was developed in the aqueous media.
In “4. Conclusion”, the novelty was emphasized as below:
This study provided a novel route for efficient conversion of corncob into FOL in a cascade reaction with solid acid Sn-SSXR and whole-cell of recombinant E. coli CF. Firstly, corncob was catalyzed with Sn-SSXR into FAL (126.3 mM) in ChCl:LA-water (30:70, v/v), and the formed FAL solution containing FAL and formate was biologically transformed into FOL with recombinant E. coli CF harboring ALDH and FDH under the mild bioreaction conditions. In addition, other biomasses were also utilized as feedstocks for the effective production of FOL. Thus, one efficient tandem conversion by bridging a chemocatalysis and a mild biocatalysis was successfully developed for sustainable conversion of biomass into FOL.
- A scheme showing the process of the material production and applications should be added.
Response: Thanks for the good suggestion. In the revised manuscript, a scheme showing the process for tandem catalysis of corncob to FOL in DES ChCl:LA-water was added.
Figure 7. Scheme for tandem catalysis of corncob to FOL in DES ChCl:LA-water
- A comparison with previously published methods and materials should be discussed and summarized in Tables.
Response: Thanks for the good suggestion. In this revised manuscript, several related works about synthesis of FAL from biomass or xylose by using tin-based solid acid were summarized in one Table. This information was given as below:
Tin-based heterogeneous catalysts (e.g., SO42−/SnO2-diatomite, SO42−/SnO2-kaoline, SO42−/SnO2-argil, SO42−/SnO2-CS, Sn-ZRD) could be utilized for efficient conversion of biomass or xylose into FAL (Table 1). Compared to Sn-SSXR, these three heterogeneous catalysts including SO42−/SnO2-kaoline, SO42−/SnO2-argil and Sn-ZRD gave lower titer of FAL. SO42−/SnO2-diatomite (3.6 wt%) could catalyze 75 g/L of corncob into FAL in a slightly high yield (68.9%) in a γ-GVL-water (6:4, v/v) system containing 15 g/L ZnCl2 at 170 ◦C after 30 min. However, high loading γ-valerolactone and ZnCl2 were used. SO42−/SnO2-CS (1.2 wt%) could dehydrate xylose (75 g/L) into 200 mM FAL in ChCl:EG-water (5:95, v/v) at 185 ◦C within 20 min. However, FAL was obtained in a low yield (35.7%). Clearly, this established catalytic process by using Sn-SSXR as chemocatalyst had potential application for FAL production.
Table 1 Related works about synthesis of FAL from biomass or xylose by using tin-based solid acid
|
DES |
|
Reaction condition |
Feedstock |
FAL concentration |
FAL yield |
Ref. |
|
SO42−/SnO2-diatomite (3.6 wt%) |
|
γ-Valerolactone-water (6:4, v/v), 15 g/L ZnCl2, 170 ◦ C, 30 min |
Corncob (75 g/L) |
- |
68.9% |
16 |
|
SO42−/SnO2-kaoline (3.5 wt%) |
|
Toluene-water (1:2 v/v), 10 mM OP-10, 170 ◦ C, 30 min |
Corncob-derived xylose (20.1 g/L) |
50.5 mM |
74.3% |
4 |
|
SO42−/SnO2-argil (3.6 wt%) |
|
Water, 180 oC, 20min |
Xylose (20 g/L) |
107.6 mM |
57.1% |
9 |
|
Sn-ZRD (3.6 wt%) |
|
Water, 170 ◦ C, 30 min |
Dewaxed corncob (75 g/L) |
90.3 mM |
- |
34 |
|
SO42−/SnO2-CS (1.2 wt%) |
|
ChCl:EG-water (5:95, v/v), 185 oC, 20 min |
Xylose (75 g/L) |
200 mM |
35.7% |
39 |
|
Sn-SSXR (2.0 wt%) |
|
ChCl:LA-water (30:70, v/v), 170 °C, 20 min |
Corncob (75 g/L) |
126.3 mM |
65.4% |
This study |
- Mass spectra of the reaction crude showing the furfural and their reduced product should be added.
Response: Thanks for the good suggestion. However, due to COVID-19 and summer holiday, experiments and tests cannot be carried out. In our work, furfural (FAL) and furfuryl alcohol (FOL) were assayed with HPLC.
- The language should be revised and typos should be corrected.
Response: Thanks for the good suggestion. In the revised manuscript, the language was revised and typos were corrected.

Reviewer 2 Report
The work entitled 'Efficient Bioreduction of Biomass-Derived Furfural into Furfuryl Alcohol using Biomass-derived Formate as Co-substrate in Deep Eutectic Solvent ChCl:LA-Water System' describes the preparation of furfuryl alcohol by bioreduction of furfural in ChCl:LA-Water System. The work is well structured and original, and the methods are adequately described. However, some more revisions need to be made.
In the introduction section, the part on DES is totally absent. The definition of DES should be added (10.1007/s10953-018-0793-1) and also the different applications. For instance, DES are used as solvents in chemical reactions (10.1039/D1GC03714E, 10.1039/d0ob02501a), extraction of polyphenols from biomass (10.1021/acssuschemeng.0c04945), delignification of biomass (10.1016/j.indcrop.2021.113692), enantiodiscrimination of chiral compounds (10.1016/j.electacta.2021.138189). In addition, the issue of water and DES should be addressed. The limit between water in DES and DES in water has been extensively studied in the literature (10.1021/acssuschemeng.9b05096, 10.1002/anie.201702486). This limit appears to be perfectly consistent with the reputed results. The authors should discuss the results in relation to what is reported in the literature concerning the role of water in DES.
In addition, the role of lactic acid in the inhibition of enzymes in cellulose saccharification has been reported in the literature. However, this inhibition does not seem to have been highlighted by the authors. This aspect should be emphasised.
One aspect that has been poorly described is that of recycling. The method of solvent recovery has not been described, nor have analyses concerning the purity of the recovered DES been shown (e.g. NMR). This aspect is of fundamental importance as the chemical stability of DES ChCl:LA has been studied in detail showing the formation of choline esters and lactide (10.1002/cssc.202002301).
Author Response
The work entitled 'Efficient Bioreduction of Biomass-Derived Furfural into Furfuryl Alcohol using Biomass-derived Formate as Co-substrate in Deep Eutectic Solvent ChCl:LA-Water System' describes the preparation of furfuryl alcohol by bioreduction of furfural in ChCl:LA-Water System. The work is well structured and original, and the methods are adequately described. However, some more revisions need to be made.
In the introduction section, the part on DES is totally absent. The definition of DES should be added (10.1007/s10953-018-0793-1) and also the different applications. For instance, DES are used as solvents in chemical reactions (10.1039/D1GC03714E, 10.1039/d0ob02501a), extraction of polyphenols from biomass (10.1021/acssuschemeng.0c04945), delignification of biomass (10.1016/j.indcrop.2021.113692), enantiodiscrimination of chiral compounds (10.1016/j.electacta.2021.138189). In addition, the issue of water and DES should be addressed. The limit between water in DES and DES in water has been extensively studied in the literature (10.1021/acssuschemeng.9b05096, 10.1002/anie.201702486). This limit appears to be perfectly consistent with the reputed results. The authors should discuss the results in relation to what is reported in the literature concerning the role of water in DES.
Response: Thanks for the good suggestion. These references were cited in the “1. Introduction” of revised manuscript. The introduction of DES was given as below:
Recently, deep eutectic solvent (DES), which is prepared by mixing a hydrogen bond donor (HBD) and a hydrogen bond acceptor (HBA) [20], has been used as green reaction solvents for the application of biocatalysis and chemocatalysis [21-23]. It is also utilized in extraction of polyphenols from biomass [24], delignification of biomass [25,26], and enantiodiscrimination of chiral compounds [27]. To reduce the dosage of DES, the mixture of DES and water has been utilized [28,29]. It has been gained great interest to utilize in the efficient production of FAL in a green water-DES system.
- Martins, M.-A.-R., Pinho, S.-P., Coutinho, J.-A.-P. Insights into the nature of eutectic and deep eutectic mixtures. J. Solution Chem. 2019, 48, 962–982
- Nolan, M.-D.; Mezzetta, A.; Guazzelli, L.; Scanlan, E.-M. Radical-mediated thiol–ene ‘click’ reactions in deep eutectic solvents for bioconjugation. Green Chem. 2022, 24, 1456-1462
- Capriati, V.; García-Álvarez, J. Copper-catalyzed Goldberg-type C–N coupling in deep eutectic solvents (DESs) and water under aerobic conditions. Org. Biomol. Chem. 2021, 19, 1773-1779.
- Panić, M.; Cvjetko Bubalo, M.; Radojčić Redovniković, I. Designing a biocatalytic process involving deep eutectic solvents. J. Chem. Tech. Biotech. 2021, 96, 14-30.
- Husanu, E.; Mero, A.; Rivera J.-G.; Mezzetta, A.; Ruiz, J.-C.; D’Andrea, F.; Pomelli, C.-S.; Guazzelli, L. Exploiting Deep Eutectic Solvents and Ionic Liquids for the Valorization of Chestnut Shell Waste. ACS Sustainable Chem. Eng. 2020, 8, 18386–18399.
- Li, W.-X.; Xiao, W.-Z.; Yang Y.-Q.; Wang, Q.; Chen, X.; Xiao, L.-P.; Sun, C.-R. Insights into bamboo delignification with acidic deep eutectic solvents pretreatment for enhanced lignin fractionation and valorization. Ind. Crop. Prod. 2021, 170, 113692
- Wu, M.; Gong, L.; Ma, C.; He, Y.-C. Enhanced enzymatic saccharification of sorghum straw by effective delignification via combined pretreatment with alkali extraction and deep eutectic solvent soaking. Bioresour. Technol. 2021, 340, 125695.
- Cicco, L.; Hernández-Fernández, J.A.; Salomone, A.; Vitale, P.; Ramos-Martín, M.; González-Sabín, J.; Soto, A.-P.; Perna, F.-M.; Arnaboldi, S.; Mezzetta, A.; Grecchi, S.; Longhi, M.; Emanuelea, E; Rizzoc, S.; Arduini, F.; Micheli, L.; Guazzelli, L.; Mussini, P.R. Natural-based chiral task-specific deep eutectic solvents: A novel, effective tool for enantiodiscrimination in electroanalysis. Electrochim. Acta 2021, 380, 138189.
- López-Sala, N.; Vicent-Luna, J.-M.; Imberti, S.; Posada, E.; Roldán, M. J.; Anta, J.-A.; Salvador R.-G.; Balestra, S.-R.-G.; Madero Castro, R.-M.; Calero, S.; Jiménez-Riobóo, R.-J.; Gutiérrez, M.-C.; Ferrer, M.-L.; del Monte, F. Looking at the “water-in-deep-eutectic-solvent” system: A dilution range for high performance eutectics. ACS Sustainable Chem. Eng. 2019, 7, 17565–17573
- Hammond, O.-S.; Bowron, D.-T.; Edler, K.-J. The effect of water upon deep eutectic solvent nanostructure: An unusual transition from ionic mixture to aqueous solution. Angew. Chem. Int. Ed. 2017, 56, 9782–9785.
In addition, the role of lactic acid in the inhibition of enzymes in cellulose saccharification has been reported in the literature. However, this inhibition does not seem to have been highlighted by the authors. This aspect should be emphasised.
Response: Thanks for the good suggestion. In this study, we didn’t investigate the effect of lactic acid on the bioreduction. In DES ChCl:LA-water (30:70, v/v), time courses for bioreduction of dilute FAL (110.0 mM) were monitored. Within 165 min, FAL concentrations dropped quickly (Figure 6). In ChCl:LA-water, no significant inhibition was observed.
Figure 6. Tandem catalysis of corncob to FOL by bridging chemocatalysis with Sn-SSXR and biocatalysis with CF cell in DES ChCl:LA-water [Arrow: pH adjustment and CF cell addition]
In the revised manuscript, the effect of ChCl:LA-water on the bioreduction of corncob-derived FAL was emphasized as below:
In ChCl:LA-water (30:70, v/v), corncob (3.0 g, 75 g/L) was tandemly converted into FOL with Sn-SSXR chemocatalyst and CF cell biocatalyst. At 180 °C, Sn-SSXR (2 wt%) converted corncob to 126.3 mM FAL in the yield of 65.4% after 15 min. This FAL liquor, which was regulated to pH 6.5, was biologically reduced to FOL with CF cells (0.1 g/mL) in ChCl:LA-water (30:70, v/v). Time courses for bioreduction of dilute FAL (110.0 mM) were monitored. Within 165 min, FAL concentrations dropped quickly (Figure 6). In ChCl:LA-water, no significant inhibition was observed. Bioreduction for 165 min, FOL concentrations reached 110.0 mM, FAL could be fully converted into FOL, achieving the productivity of 0.66 g FOL/g xylan.
One aspect that has been poorly described is that of recycling. The method of solvent recovery has not been described, nor have analyses concerning the purity of the recovered DES been shown (e.g. NMR). This aspect is of fundamental importance as the chemical stability of DES ChCl:LA has been studied in detail showing the formation of choline esters and lactide (10.1002/cssc.202002301).
Response: Thanks for the good suggestion. The recovery and reuse of reaction medium ChCl:LA-water and catalyst Sn-SSXR were conducted as below:
The reusability of chemocatalysts and reaction media has a vital role in the cost-effective manufacture of FAL [29,30]. After each reaction, FAL was extracted from FAL liquor for three times using ethyl acetate by mixing in equal volume. Furthermore, the reaction medium ChCl:LA-water was separated from solid samples (Sn-SSXR and biomass residues). After the solid samples were calcined in muffle furnace (550 °C) for 4 h, and the obtained heterogeneous catalyst regeneration was carried out by sulfonating the recycled catalysts before each batch of use. In this work, recovered and repeated reuse of heterogeneous Sn-SSXR and DES ChCl:LA was carried out for six batches. As depicted in Figure 3, FAL yield reached 65.4% in 1st batch. In the 2nd and 3rd run, slight decrease in FAL yield was found. From 4th to 6th run, FAL yield decreased from 46.3% to 34.1% The reused Sn-SSXR and DES ChCl:LA-water had high stability for the conversion of biomass to FAL in ChCl:LA-water. The recovery and reuse of the catalytic system could relieve economic burden and reduce environmental pollution, and has a high application potential.
In this work, the reaction medium DES ChCl:LA-water was recovered and reused. The DES was not further purified. In addition, due to COVID-19 and summer holiday, experiments and tests for assaying the purity of the recovered DES (e.g. NMR) cannot be carried out.
This reference was cited in this revised manuscript.
- Morais, E.-S.; Lopes, A.-M.-D.-C.; Freire, M.-G.; Freire, C.-S.-R.; Silvestre A.-J.-D. Unveiling modifications of biomass polysaccharides during thermal treatment in cholinium chloride:lactic acid deep eutectic solvent. ChemSusChem 2021, 21, 686-698.

Round 2
Reviewer 1 Report
The authors addressed most of the comments and the revised version can be accepted.
Author Response
Reviewer 1#
The authors addressed most of the comments and the revised version can be accepted.
Response: Thanks for this good comment.
Reviewer 2 Report
I would like to thank the authors for accepting my suggestions. The changes made are satisfactory. Given that it was not possible to do analysis on the recycled medium, I recommend that a sentence be included where the study of recycling is deferred to later work. The issue of DES recycling is a key factor in green chemistry. In added the authors' names in reference 27 are incorrect. The first 8 authors must be removed. Finally, I strongly suggest that an outline of the synthetic pathway be included so as to make it clearer what reactions are being studied. After these minor revisions are made, the work is suitable for publication.
Author Response
Response Letter
Reviewer 2#
I would like to thank the authors for accepting my suggestions. The changes made are satisfactory. Given that it was not possible to do analysis on the recycled medium, I recommend that a sentence be included where the study of recycling is deferred to later work. The issue of DES recycling is a key factor in green chemistry. In added the authors' names in reference 27 are incorrect. The first 8 authors must be removed. Finally, I strongly suggest that an outline of the synthetic pathway be included so as to make it clearer what reactions are being studied. After these minor revisions are made, the work is suitable for publication.
Response: Thanks for the good suggestion. In this revised manuscript, some revisions were given as below:
1) Given that it was not possible to do analysis on the recycled medium, a sentence was included as following:
The reusability of chemocatalysts and reaction media has a vital role in the cost-effective manufacture of FAL [38,39]. After each reaction, FAL was extracted from FAL liquor for three times using ethyl acetate by mixing in equal volume. Furthermore, the reaction medium ChCl:LA-water was separated from solid samples (Sn-SSXR and biomass residues). After the solid samples were calcined in muffle furnace (550 °C) for 4 h, and the obtained heterogeneous catalyst regeneration was carried out by sulfonating the recycled catalysts before each batch of use. In this work, recovered and repeated reuse of heterogeneous Sn-SSXR and DES ChCl:LA was carried out for six batches. As depicted in Figure 3, FAL yield reached 65.4% in 1st batch. In the 2nd and 3rd run, slight decrease in FAL yield was found. From 4th to 6th run, FAL yield decreased from 46.3% to 34.1% The reused Sn-SSXR and DES ChCl:LA-water had high stability for the conversion of biomass to FAL in ChCl:LA-water. The recovery and reuse of the catalytic system could relieve economic burden and reduce environmental pollution, and has a high application potential. The issue of DES recycling is a key factor in green chemistry. An efficient strategy for recovery and reuse of DES ChCl:LA in under progress.
2) In reference 27, some author names are incorrect. The first 8 authors were removed. Reference [27] was revised as following:
“27. Cicco, L.; Hernández-Fernández, J.A.; Salomone, A.; Vitale, P.; Ramos-Martín, M.; González-Sabín, J.; Soto, A.-P.; Perna, F.-M.; Arnaboldi, S.; Mezzetta, A.; Grecchi, S.; Longhi, M.; Emanuelea, E; Rizzoc, S.; Arduini, F.; Micheli, L.; Guazzelli, L.; Mussini, P.R. Natural-based chiral task-specific deep eutectic solvents: A novel, effective tool for enantiodiscrimination in electroanalysis. Electrochim. Acta 2021, 380, 138189.” was revised to “27. Arnaboldi, S.; Mezzetta, A.; Grecchi, S.; Longhi, M.; Emanuelea, E; Rizzoc, S.; Arduini, F.; Micheli, L.; Guazzelli, L.; Mussini, P.R. Natural-based chiral task-specific deep eutectic solvents: A novel, effective tool for enantiodiscrimination in electroanalysis. Electrochim. Acta 2021, 380, 138189”.
3) An outline of the synthetic pathway was included so as to make it clearer what reactions are being studied. It was given as below:
Scheme 1. Outline of the synthetic pathway.
